# Canada’s Colonial Genocide of Indigenous Peoples: A Review of the Psychosocial and Neurobiological Processes Linking Trauma and Intergenerational Outcomes

**DOI:** 10.3390/ijerph19116455

**Published:** 2022-05-26

**Authors:** Kimberly Matheson, Ann Seymour, Jyllenna Landry, Katelyn Ventura, Emily Arsenault, Hymie Anisman

**Affiliations:** 1Department of Neuroscience, Carleton University, Ottawa, ON K1S 5B6, Canada; jyllennalandry@cmail.carleton.ca (J.L.); katelynventura@cmail.carleton.ca (K.V.); emilyarsenault@cmail.carleton.ca (E.A.); hymieanisman@cunet.carleton.ca (H.A.); 2University of Ottawa Institute of Mental Health Research at the Royal, Ottawa, ON K1Z 7K4, Canada; 3School of Social Work, Carleton University, Ottawa, ON K1S 5B6, Canada; annseymour3@cmail.carleton.ca

**Keywords:** Indigenous, colonization, historical trauma, intergenerational, early-life adversity, epigenetics, mental health, neurobiological, determinants of health

## Abstract

The policies and actions that were enacted to colonize Indigenous Peoples in Canada have been described as constituting cultural genocide. When one considers the long-term consequences from the perspective of the social and environmental determinants of health framework, the impacts of such policies on the physical and mental health of Indigenous Peoples go well beyond cultural loss. This paper addresses the impacts of key historical and current Canadian federal policies in relation to the health and well-being of Indigenous Peoples. Far from constituting a mere lesson in history, the connections between colonialist policies and actions on present-day outcomes are evaluated in terms of transgenerational and intergenerational transmission processes, including psychosocial, developmental, environmental, and neurobiological mechanisms and trauma responses. In addition, while colonialist policies have created adverse living conditions for Indigenous Peoples, resilience and the perseverance of many aspects of culture may be maintained through intergenerational processes.

## 1. Introduction

‘*Our object is to continue until there is not a single Indian in Canada that has not been absorbed into the body politic*.’ Duncan Campbell Scott, 1920 [1]

‘*Yes, apocalypse. We’ve had that over and over. But we always survived. We’re still here.*’ Waubgeshig Rice, *Moon of the Crusted Snow* [2]

We often look to the past to learn about the impact of human atrocities, and of the collective trauma experienced by groups that have occurred across many generations (i.e., collective historical trauma). In the case of some groups (e.g., the attempted genocide of Jews during the Holocaust) considerable data have been collected showing the mental and physical health consequences of these experiences. Other instances, such as the wartime incarceration of Japanese Americans, have received far less attention. The evaluation of the Armenian genocide in which Ottoman (Turkey today) leaders were responsible for the deaths of more than a million people during World War I has hardly been acknowledged, let alone evaluated [3]. Data concerning the war in Bosnia and the mass killing of Tutsis by Hutus in Rwanda in 1994 are only now beginning to emerge [4]. Intergenerational consequences have been reported among the offspring of those who survived the Holodomor genocide of 1932–1933 when Ukraine experienced a famine instigated by Russia [5]. We can only guess the consequences of Putin’s current brutal attack on the people of Ukraine.

It is impossible and inappropriate to compare collective traumas that have been experienced by different groups, especially given the pronounced differences that exist with respect to the magnitude, strategy, and duration of the events. Distinctions in the malign motivations that prompted these atrocities require consideration. Whereas some collective assaults were aimed at eradicating all people of specific groups by killing them, in other instances, the apparent goals focused on cultural genocide that comprised the dispossession of homelands and spiritual and cultural destruction, promoting the disappearance of a people over time. Indigenous Peoples worldwide, including in Canada and the U.S. have, over generations, experienced assaults on their very survival, along with their culture, land, and ways of life. The Truth and Reconciliation Commission of Canada, established to bring to light the experiences of survivors of the Indian Residential Schools, referred to this program as constituting cultural genocide [1]. Irrespective of the many differences in how genocide plays out, these events all inflicted severe and lasting harm to the survivors. The cumulative consequences of the historical traumas experienced by Indigenous Peoples have been described as causing a ‘soul wound’ that has profoundly affected individuals across generations [6].

Some have maintained that the experiences of Indigenous Peoples in Canada constitute more than just ‘cultural’ [7]. The term genocide was first introduced by Raphael Lemkin, who relentlessly pressured the United Nations to enact laws that recognized genocide as an international crime [8]. Particularly relevant to the discussion that follows is Article II of the 1948 ‘Convention on the Prevention and Punishment of the Crime of Genocide’:

Article II: In the present Convention, genocide means any of the following acts committed with intent to destroy, in whole or in part, a national, ethnical, racial or religious group, as such:(a)Killing members of the group;(b)Causing serious bodily or mental harm to members of the group;(c)Deliberately inflicting on the group conditions of life calculated to bring about its physical destruction in whole or in part;(d)Imposing measures intended to prevent births within the group;(e)Forcibly transferring children of the group to another group.

While there has been a focus on direct acts of genocide, less often has consideration been given to the indirect impacts involving trauma, poverty, food scarcity, forced migration, loss of homes and land, loss of cultural values and overall impacts on health and well-being. Yet, if genocide constitutes bodily or mental harm to members of a group, these impacts constitute well-documented social and environmental determinants of health. In this regard, numerous legislative policies in Canada have served to limit the capacity for Indigenous Peoples to maintain their relationship and connections to the land for physical and spiritual sustenance, and disrupted the transmission of knowledge and shared ways of life across generations to ultimately undermine health and well-being. The lands of First Nations were taken, and through the unilateral creation of paternalistic federal policies such as the Indian Act, First Nations Peoples were confined to reserves located in federally demarcated areas of their ancestral lands (typically areas that were viewed by the colonizers as lacking resources of interest). Treaties were negotiated that many First Nations Peoples understood to be about sharing resources (as the notion of land ownership was not part of Indigenous world views). These were acted on by the colonizers as though the lands had been surrendered. 

The ongoing systemic neglect experienced by Indigenous Peoples has resulted in poor housing conditions, food insecurity, and the absence of potable water in some reserves. The bodily and mental harm of Indigenous Peoples was caused by policies that undermined birthing rights and maternal health, including the forcible removal of children. These historical and current conditions might further constitute acts of genocide by ‘deliberately inflicting on the group conditions of life calculated to bring about its physical destruction in whole or in part’ (Article IIc) by means of environmental, social, and cultural degradation. 

In this paper, we review the intergenerational trauma that was instigated by the historical genocide and ongoing genocidal conditions that shape the lives of Indigenous Peoples in Canada. We are not legal scholars, but rather our analysis considers these actions from a psychosocial and developmental perspective, together with the biological sequelae of these policies and actions. Grounded by the determinants of health perspectives, as shown in Figure 1, we have organized our analysis to consider the macrolevel policies of the federal government, and the implications of these policies that affected key aspects of the cultural and social environment of Indigenous Peoples in Canada. This environment is proposed to have given rise to multiple processes that contributed to the transmission of trauma across generations. In our analysis, we address how these factors represent potential mechanisms that link the effects of colonization to well-being outcomes across generations. Despite devasting transgenerational outcomes, these same processes can serve as a basis for collective resilience, and we address the sources of strength that enable adaptation, perseverance, and resistance among Indigenous Peoples, despite the continued adverse conditions.

The research literature included in this review is selective, as in many instances direct research based on human experience, let alone the experiences of Indigenous Peoples, is lacking. Thus, to demonstrate the validity of particular mechanistic processes, at times we relied on research with animals (primarily rodents), and other human populations that were subjected to collective trauma. Where available, research conducted with Indigenous populations is described. We note in each section the limits to the existing research. 

## 2. Historical Trauma: 500 Years of Colonization

Colonization has long been a way for some nations to gain wealth and influence. The African continent has experienced colonization for more than two millennia. The colonization of what was to become the Americas began in the early part of the 16th century, before which Indigenous Peoples had their own kinship and governance structures, ceremonies and belief systems, as well as knowledge developed over many generations regarding a balanced co-dependent relationship with the land. A degree of cooperation initially existed between Indigenous Peoples and the Europeans who arrived, including the sharing of skills and knowledge, as well as trade and military alliances. Thus, at the outset, relationships with Indigenous Peoples were greatly valued—until they were no longer needed and viewed as an encumbrance. As settler populations increased, their desire for land and wealth grew. The colonization of Indigenous Peoples over several centuries was achieved by guns and by old-world bioweapons that comprised the introduction of smallpox, tuberculosis, and influenza that killed more people than settler bullets did. In fact, in what is now Canada, the Indigenous population pre-colonization was estimated to have been between 350,000 and 500,000 (but some estimates have it as high as 2 million), but declined to just 125,000 by 1867.

In addition to the physical appropriation of land was the colonial effort to eliminate the transmission of cultural identity, traditional skills, and connection to the land. Beginning in 1883 (while this was the date of the first federally established church school, similar institutions existed as early as the 1830s, years before Canadian federation) Indian Residential Schools (IRSs) were established in Canada (as were American Indian Boarding Schools in 1862). Children were forcibly removed from their families and were institutionalized in IRSs with the explicit goal of ‘taking the Indian out of the child’. These mandated church-run IRSs endeavoured to save the souls of the ‘savages’ by immersing them in Euro–Christian beliefs and eradicating access to traditional socialization values, language, practices and ways of life. By the 1930s, roughly 75% of First Nations children attended IRSs, as did many Métis and Inuit children. The last of the IRSs was closed in 1996, but by then several generations of children had experienced the mistreatment that abounded in these institutions.

While the treatment of children varied across the different IRSs, they frequently experienced harsh and denigrating conditions [1]. In addition to physical, emotional and sexual abuse and neglect, children were stripped of their identities, made to feel ashamed of themselves and their culture, and denied the use of their language, beliefs, and ways of being. Some IRSs were associated with Indian hospitals where sick children were transferred but, owing to poor isolation procedures, diseases such as tuberculosis were often acquired. Indian hospitals were not intended to administer treatment that incorporated traditional medicines, midwives or holistic strategies; rather, the hospitals’ assimilationist goals were to replace traditional healing with Western biomedical procedures [9]. In the Indian hospitals, which treated Indigenous patients in addition to IRS students, non-consensual sterilization was commonly undertaken (such non-consensual operations were performed on Indigenous women even as recently as 2018). In some IRSs, nutritional experiments were conducted from 1942 to 1952 to determine whether vitamin supplements (riboflavin, thiamine, niacin, and bone meal) would limit the effects of purposefully introduced malnourishment. These experiments continued despite the deteriorating health and increased mortality of children, and the developmental delays that were produced [10]. Indeed, these experiments continued well after the UN Convention on genocide was signed. The consequences for IRS survivors resulted in marked and varied repercussions on health, which has been elucidated in detail in a tragically poignant report [11]. 

The trauma did not end for children once they were released from the IRSs. They were returned to reserves that had been subjugated into poverty, and where they felt disconnected from their families and communities. There was much discord between community members, often instigated by prior conflicts (bullying) brought about by a need to survive in the IRSs [12]. Moreover, on some reserves, divisions existed between members who had internalized the religious precepts imposed on them at the IRSs, whereas others were intent on maintaining traditional Indigenous ways of life. Thus, where communities had operated as cohesive collectives, the IRS experience fostered divisions and conflicts over fundamental values. Further contributing to dysfunction in some communities, adult survivors frequently experienced shame and guilt owing to their powerlessness in protecting their children or siblings from the IRS experience, resulting in coping behaviors that were self-abusive or abusive of others. 

Although the experiences of the children who attended the IRSs and the consequences that emerged are relatively well known, far less attention has been directed toward the Sixties Scoop. During the period from the 1950s to the 1990s, about 20,000 Indigenous children in Canada were removed from their families and communities by child welfare services. Indigenous children whose parents attended IRSs were at greater risk of experiencing childhood adversity in their homes, thus increasing the likelihood of their removal and placement with non-Indigenous foster parents, adoptive parents, and in group homes, often permanently separated from their families and from their cultural identity [13,14]. 

Even today, despite representing only 7.7% of the Canadian population, Indigenous children represent over half (52.2%) of children aged 14 and younger in foster care, described as the ‘millennial scoop’ [15]. Remarkably, there are currently more Indigenous children in the care of the child welfare system (CWS) than there were in IRSs at any single point in time [1]. In numerous regional jurisdictions in Canada, there remains a policy of putting expecting mothers, disproportionately those who are Indigenous, on ‘birth alerts’, with social workers being present to apprehend a newborn infant immediately following the birth without parental consent. Even in regions where birth alerts are no longer official policy, it has been suggested that the practice continues, largely due to systemic biases within the CWS. There is growing evidence that the overrepresentation of Indigenous children at risk of apprehension by the CWS can be ascribed to social and economic inequities [13,16,17], most of which are linked to the continued legacy of governmental forced assimilation policies [13]. Sadly, experiencing the child welfare system is predictive of poorer mental health outcomes, along with juvenile and adult incarceration in the criminal justice system. In addition to affecting the children taken, these experiences promoted profound mental health consequences in parents [18]. 

Colonialist actions involving policies such as the IRSs and the Sixties Scoop, along with forced relocations and the restricted movement of Indigenous community members, have contributed to the erosion of the knowledge of the land that was passed down through generations. Treaties were created that were regarded by settlers to represent a surrender of the land by First Nations. The Indian Act restricted First Nations Peoples to reserves and diminished access to the broad resources of their ancestral territories either by limiting practices, physical access, or by developments that destroyed the land, contaminated the water, or eradicated traditional ways of life and livelihoods. As with so many Indigenous populations globally, the land is not perceived as an object to be possessed. Instead, the plants, animals, and waters are viewed as ‘life’ and inter-connected relationships, as ‘members of the larger other-than-human community’ [19]. Thus, the systematic removal of Indigenous Peoples from their lands and the development (desecration) of traditional territories, ranging from clear-cutting to changes to waterways (hydro dams) to fracking and pipelines, to the toxic disposal of pollutants in the air and water systems, all constitute traumatic insults with incumbent consequences [20]. The Canadian determination to establish sovereignty in remote areas (e.g., the Arctic circle) involved the duplicitous relocation of Inuit families to thousands of kilometres from their home territories. Elders have recounted decreased physical, mental, and emotional health after experiencing dispossession from the land [21]. Spiritual impact and loss are felt related to the inability to pass on traditional Indigenous knowledge to younger generations; reduced access to the land preventing its use for ceremonies, traditional practices and sustenance; the reduced ability for water to give and support life; the loss of language that is essential to express their connection with; and lost knowledge of the land [21,22]. 

Beyond colonial exploitation of the land, environmental racism and climate change are impacting the health and well-being of Indigenous Peoples, especially in the north [23,24]. The most recent report of the Intergovernmental Panel on Climate Change [25] points to climate change as a contemporary manifestation of colonialism. In this regard, colonialism is implicated as both a driver of climate change and in exacerbating the vulnerability of Indigenous communities. When the land is the main source of sustenance, the inability to predict environmental phenomena or the lack of tools to contend with such changes can have particularly devastating effects on well-being at both the individual and community level. Many Indigenous communities have been contending with critical environmental issues, including the toxicity of local bodies of water and exposure to natural and human-made disasters, such as floods and forest fires. When such events force displacement, temporarily or permanently, there is the additional diminishment of the safety and security of displaced populations. Under these conditions, resources (e.g., financial, along with basic needs such as water, food, and clean air) are stretched or no longer accessible, family and community members may be divided or lost, safety and security are undermined, and exposure to potential toxicants and disease are heightened. Warming temperatures, as well as changes in weather patterns and ice stability, have altered access to resources critical to community survival and created barriers to community members’ participation in land-based practices and ways of life, resulting in diminished cultural identity and well-being [23,26]. These effects are occurring even though Indigenous Peoples’ ways of life are least likely to be contributing to the human-made impact on the climate. On the contrary, many of the environmental disasters that have affected the lives and livelihoods of Indigenous Peoples in Canada have, at their roots, industrial developments by corporations with no or minimal Indigenous representation, consultation or benefit. 

The built environment of Indigenous communities may further promote severe health issues. Government programs that control the building of homes and infrastructure in fly-in and northern communities apply standards that are not only inappropriate for the climate [27,28], but disregard the cultural and social needs associated with familial and community living. Homes were built with standard designs defined by government agencies (‘Indian Houses Type 1–7’), all with similar floor plans. This strategy undermines traditional family structures and diminishes individuals’ identification with their living spaces [27,28]. The houses in these communities are often built below acceptable quality standards, and without consideration of the high cost of transporting materials to remote communities, thereby limiting the number of new homes and extending the time needed to build. 

A failure to include Indigenous perspectives in decision-making continues to severely limit progress for self-determination in housing [28]. The lack of reconciliation in housing policies combined with the substandard construction of homes on-reserve has resulted in statistics that have remained unchanged for decades; in 2006, 44% of on-reserve First Nations homes required major renovations and 36% reported living in overcrowded conditions, and these numbers were still evident in 2016 [29,30]. It has been noted that, ‘The consequence of this crisis is that the home, which for First Nations people has traditionally been a place of pride and identity, instead today exacerbates many social and health problems’ [27] (p. 2). Some of these social problems contribute to intergenerational trauma, including numbing the pain by substance misuse and other related negative coping mechanisms [31].

Colonization as it is recapitulated in current times meets the key criteria of Article II of the UN Convention on Genocide. The intergenerational effects of historic and ongoing systemic racism have had considerable implications for the health and well-being of Indigenous Peoples (Article IIb). Mental health problems are endemic among Indigenous populations; depressive disorders, anxiety, PTSD, and substance use disorders occur at inordinately high levels. In many First Nations and Inuit communities, the occurrence of suicide has been much higher than in the remainder of the Canadian population. The frequency of suicide varies across communities, ranging from 2 to 10 times that of non-Indigenous people, and it is especially disconcerting that over the last two decades suicide has occurred more frequently among young people. This is similarly apparent among American Indians and Alaskan Natives, in whom suicide occurs at 2.5 times the frequency of the population at large. 

Understandably, a significant level of distrust between Indigenous Peoples and healthcare professionals exists. Given women’s experiences of forced sterilization, birth alerts, and the removal of children as a result of various policies over generations, the fear of child apprehension is so deeply rooted that some mothers may refuse skin-to-skin contact, breastfeeding, and other bonding strategies because they expect their child to be taken from them (Articles IId, IIe). This learned distrust of healthcare professionals jeopardizes the willingness of expectant mothers to disclose their medical, physical, mental or substance use history. Such mistrust is particularly evident among Indigenous women in fly-in and northern communities. The medical and healthcare services offered in such communities are often limited, forcing expectant mothers to leave the community to deliver in larger, urban, mainstream hospital facilities, and those who experience pregnancy complications are sometimes away from home for extended periods of time. Forced obstetric evacuation is particularly stressful, as due to the nature of medical funding that is applied to Indigenous Peoples, women are typically separated from family and community support for the final weeks of pregnancy, putting a notable strain on the health and wellness of both mother and child. This may contribute to the increased risk of medical intervention during labor, as well as postpartum depression, along with the negative developmental consequences of stress on the fetus, and may even account for why the mortality rate of Indigenous infants is 2–7 times higher than non-Indigenous infants [32]. The loss of the birthing experience and related ceremonies in communities is perceived as a cultural loss and forced evacuation has been long associated with colonial practices. 

Just as colonialism continues to influence Indigenous entrance into the world, it also influences death. The built environment hinders the palliative care that reflects the values of Indigenous Peoples. A lack of healthcare access on reserves and inadequate housing that does not allow for proper care necessitates aging individuals leaving their community to access care in a hospital setting [29,33,34]. While the majority of Indigenous people would prefer to pass away at home, many die in urban hospitals where the physical structure of the hospital building or prohibitive administrative policies can prevent end-of-life care that aligns with Indigenous values. In addition, this stage of life is often a time of sharing stories and passing on knowledge, and so death in urban hospitals is another form of systemic racism that further marginalizes Indigenous ways of knowing [29]. 

## 3. Examining the Intergenerational Effects of Trauma: The Mental and Physical Harm Perpetrated against Indigenous Peoples

In the next section of this paper, we will provide an understanding of how the diminished mental and physical health outcomes associated with colonization occur not only among the survivors of historical and ongoing traumas such as IRSs and involvement in the child welfare system, but how these experiences are felt across generations, including from parent to child (intergenerational) as well the historic trauma that affects multiple generations (transgenerational effects). It is, in part, for this reason, that the acts perpetrated as a result of the colonialist practices and policies ought to be viewed as constituting genocide. Even those actions that did not directly comprise ‘killing members of the group’ (although the recent uncovering of unmarked mass burial sites of children who died in the IRSs suggests otherwise), the transgenerational impacts on survivors have undermined the health expectancy and life expectancy of several generations. The heightened impact of stressors may be especially pronounced when superimposed on a background of ongoing distress, such as poverty, which itself may have arisen owing to the trauma experienced in an earlier generation, or from toxic living conditions. The social and financial consequences that emanated from earlier trauma as well as the psychological damage that had been inflicted on the parents may have hampered their parenting abilities, thereby contributing to the transmission of trauma effects [35]. Finally, dispossession from the land disrupts access to a way of life and life-sustaining resources, and undermines the capacity to cope with trauma in a way that might facilitate healing processes.

### 3.1. Impact of Early-Life Stressors

Stressful experiences at any stage of life can promote adverse health repercussions, but those encountered early in life may engender especially damaging effects. If parents encounter traumatic or chronic stressors or have mental health problems, including substance use disorders, their parenting abilities may be undermined. The poor parenting associated with distressing events may comprise disturbed parent–child interactions, neglectful behaviors, or disengagement from children, as well as hostility, coercion, or abusive behaviors, which may result in the child’s psychological development being disturbed. 

Young children typically have not developed effective coping strategies and their restricted social connections limit the help they can acquire from others [36]. Thus, adverse childhood experiences (ACEs) may disrupt school performance, foster the mistrust of others, undermine the ability to form and maintain close relationships, and self-regulation may be disturbed, culminating in the emergence of psychological disorders [37]. In the context of parental mistreatment, children may form self-damaging inferential attributions concerning the abusive or neglectful experiences happening to them. They may internalize the belief that these adverse events are justified, and attributable to aspects of themselves (i.e., self-blaming and self-criticizing). ACEs have been associated with an elevated risk of subsequent stressor encounters (stress proliferation) as well as revictimization (e.g., domestic abuse) [38]. The distress related to a sense of threat, as well as the uncertainty and unpredictability concerning future harm being experienced, may increase vulnerability to the development of anxiety and depression and may be a driving force in the development of PTSD following a subsequent traumatic experience [39]. The more frequently ACEs are encountered, the more likely it is that adult psychopathology will develop [40]. 

To a significant degree, cognitive disturbances linked to multiple brain changes act as mediators between early experiences and the emergence of psychopathology [41]. These features may become psychologically embedded and may carry through to adulthood; thus, cognitive distortions may develop such as a preoccupation with danger or exaggerated concerns of future harm [42], with these even being evident during old age [43]. These experiences are likewise more likely to disturb immune and inflammatory processes, thereby increasing the risk of numerous non-communicable illnesses, such as immune-related disorders, type 2 diabetes, and heart disease [44].

The present-day over-representation of Indigenous children in the child welfare system in Canada, including those apprehended at birth, is purported to reflect ongoing paternalistic attitudes and policies that perpetuate and interact with the long-term consequences of the IRS system and the Sixties Scoop [13]. The sudden separation of pre-adolescent children from their parents can produce marked adverse effects, as witnessed among immigrants attempting to enter the US from Mexico [45]. It has been known for decades that raising infants within orphanages (within Romania) where contact comfort is limited may have multiple adverse developmental consequences [46], including pronounced brain electrophysiological alterations [47]. Indeed, the absence of skin-to-skin contact might have negative repercussions in infants, although the data within most studies comprise only a small number of participants and vary across the specific outcomes assessed [48]. The case has been made that skin-to-skin contact is essential for the development of the neurohormonal changes that limit stress reactivity. The absence of such contact during the initial hours following birth has been related to poor mother–infant interactions assessed one year later [49]. Mothers separated from children may experience lasting grief and the frequency of postpartum depression may be elevated [50]. 

Damaging effects of parental trauma may occur even in relatively well-adjusted families owing to the communication patterns between parents and their children. Having experienced severe trauma, such as among some survivors of the Holocaust or the IRSs, there may be a need to unburden themselves of the horrors they have experienced [51]. Among others, in contrast, there is a tendency to withhold any communication about their experiences, which has been dubbed a ‘conspiracy of silence’ [52], furthering their feelings of shame and isolation [53]. In fact, in many cases, the children of IRS survivors did not know that their parents had attended one of these institutions; it was not until the IRS Settlement Agreement in 2005 (which offered reparations to survivors) or the 2008 IRS apology from the government that many survivors began talking to their adult children [54]. In an astute analysis concerning the impact of the lack of communication among survivors of the Holocaust, Hirsch [55] described the creation of a ‘postmemory’ or a ‘reclaiming of memory’ in which children of survivors stitched together bits of information that they obtained, recreating narratives and images to form a memory of these experiences. Hirsch described these as ‘so powerful, so monumental, as to constitute memories in their own right’ [55] (p. 16). As much as Holocaust survivors might have wished to protect their children, those who experienced the silence were, unfortunately, more likely to be vulnerable to experiencing interpersonal distress [56]. 

It has been known for decades that children who experience poor parenting may subsequently model their own child-rearing behaviors based on their experiences: ‘poor parenting begets later poor parenting’ [35]. It has been estimated that child maltreatment occurs in about 30% of children whose parents have themselves been maltreated. Thus, the transmission of poor parenting may stem from abuse experienced by parents during their own childhood [57], adversity related to poverty, diverse traumas encountered, or impaired mental health [58]. Given the ‘parenting’ experienced by survivors of the IRSs, it is no wonder that their capacity to parent was undermined simply by virtue of the role models that they were provided. An extensive analysis that focused on the impacts of the IRSs revealed that relatively poor quality of parenting was reported by Indigenous youth who had one or both parents attend an IRS, and this was similarly experienced by individuals whose grandparents had attended Residential Schools. Moreover, First Nations adults who had a parent who attended IRS reported experiencing relatively frequent adverse childhood experiences and adult traumas. These individuals self-reported greater depressive symptoms and thoughts of suicide in comparison to adults whose parents did not attend an IRS. These psychological disturbances were evident in third-generation survivors and were particularly notable among individuals who had both a parent and grandparent attend an IRS [59]. 

All of the studies conducted looking at the processes associated with the intergenerational transmission of trauma are correlational using observational or self-reported data. Such data limit conclusions of causality, as many factors might contribute to effects over time and generations. Although studies in animals are not always generalizable to humans, it is nevertheless significant that research on multiple species has found that the behavior of parents toward their offspring can have intergenerational and transgenerational consequences. For example, numerous studies in rodents have demonstrated that, among pups separated from their mothers for short periods, maternal behaviors were altered and anxiety and erratic behaviors were subsequently apparent in the offspring [60,61].

### 3.2. Living Conditions: The Built Environment

Housing is a determinant of health, with poor housing being linked to a variety of negative health outcomes, including injuries and disease [62,63]. As well, inadequate housing is associated with psychological and social challenges, either as a direct cause or as an exacerbating factor of illness [63]. A clear example can be seen with tuberculosis. Though Canada currently ranks among the lowest rates of tuberculosis in the world, the incidence rate among Indigenous Peoples, particularly among Inuit, is 290 times higher than the Canadian non-Indigenous population [64]. This has been linked to the severe housing shortages in Nunavut [65], where over half (52%) of the population lives in social housing [66]. As much as 72% of social housing tenants live in housing that is overcrowded; frequently, 20 people may be living in a four-bedroom home [65]. Crowding combined with mould and other housing inadequacies creates an ideal environment for the development and transmission of tuberculosis and other respiratory diseases.

Crowding itself is not necessarily the primary issue, as extended family living in single domiciles is not uncommon [28]. The problem fundamentally comes down to the lack of choice in housing design and fit for cultural and climatic conditions, along with siloed funding and housing development programs that are not community-led and fail to consider Indigenous family values [27,28]. This said, crowding can still contribute to psychological distress, particularly in the presence of poor parenting practices related to prior trauma [28,67,68]. Housing shortages that impose multigenerational cohabitation can enable the perpetuation of cycles of abuse [67,69]. Additionally, the prevalence of dilapidated housing functions as a direct reminder of the continued influence of colonial policies and forced relocation, which does not create an environment conducive to emotional healing [27]. 

Housing inadequacy contributes to children being taken from their homes at alarmingly high rates. Appearances of unsafe housing increase the risk of a family being investigated by child welfare services, even after controlling for other psychosocial and socioeconomic variables [69]. While the study by Hirsch and colleagues [69] did not specifically look at children from Indigenous families, other researchers have found that Indigenous children are more likely to be placed in care outside of their home due to housing difficulties compared with non-Indigenous children [70]. Furthermore, in some instances, the situation is exacerbated when a parent attempts to address their own mental health issues in an effort to create a better home environment for their child [71,72]. For example, if a parent enters a live-in substance rehabilitation program, this entails leaving children alone or in the care of others who may or may not act in the child’s best interests. The parent may need to take time away from work which, due to diminished income, may cause further distress, including the possibility of eviction. A parent may be required to meet specific housing requirements based on Western standards, such as separate bedrooms for each child, in order to maintain custody of their children [73]. The severe housing shortage in communities can present a serious barrier for meeting child welfare system standards. Social assistance programs further contribute to housing precariousness when a child is taken into the child welfare system, as the family may lose the financial assistance that was helping them to pay for their home [73]. In the Northwest Territories specifically, a single parent living in social housing will be evicted from their home if they lose custody of their child, and there is limited housing available for single adults with no children [74]. These examples of government-mandated requirements that intensify the challenges families are already experiencing have been identified as institutional racism [72]. It has been suggested that ‘institutional racism severed many First Nations’ from their inherent right to the traditions and values’ of culture and identity [75] (p. 84). Housing represents a key opportunity to address multiple pathways for reconciliation and cultural safety, as the home functions as a central point of convergence for the detrimental psychological, social, and environmental impacts of colonial genocide as well as the connections to family and community that are critical for resilience [74,76,77]. 

## 4. Neurobiological Consequences of Early-Life Trauma

Aside from the psychosocial and environmental factors associated with diminished outcomes emanating from traumatic experiences, numerous biological consequences of chronic and traumatic stressors can contribute to varied psychological and physical illnesses. These studies, primarily conducted in animal models, have indicated that stressful experiences profoundly affect numerous hormonal processes (e.g., CRH, ACTH, cortisol, estrogen, oxytocin, arginine vasopressin), brain neurotransmitters (e.g., monoamines, GABA, glutamate), neurotrophins (e.g., BDNF, FGF-2, VEGF), brain microglia functioning, and gut microbiota, as well as immune and inflammatory responses that may promote or exacerbate physical and psychological disturbances [44,78,79]. To a considerable extent, many of these effects are moderated by genetic and epigenetic factors, the organism’s previous stressor experiences, and a constellation of psychosocial and environmental variables [80].

The neurobiological changes introduced by stressors are particularly pronounced when encountered early in life, varying with the nature and intensity of the stressors experienced. If early-life stressors are not intense (tolerable stressors), these experiences may teach children how to cope, and could thus serve in a protective capacity [81]. Severe adverse events (sometimes referred to as toxic stressors), in contrast, may result in the sensitization of biological processes so that the later reactions to diverse stressors may cause exaggerated behavioral and biological responses [80]. 

In addition to sensitized neurobiological responses, if stressor experiences are sufficiently protracted, biological systems (e.g., immune functioning) can be overly taxed, thereby influencing vulnerability to immune and inflammatory disorders [44]. Likewise, chronic stressor exposure may promote hippocampal receptor loss owing to their sustained activation by glucocorticoids (allostatic overload), potentially influencing cognitive abilities [82]. From this perspective, early life stressful experiences may prime biological systems so that allostatic overload will be provoked more readily, thereby promoting specific disturbances. It is equally possible that the broad actions of adverse early-life events may engender a general susceptibility so that the risk of multiple physical and mental illnesses is elevated [83]. 

As informative as studies in rodents are in documenting the influence of chronic stressors and determining the processes governing these actions, their relevance in informing trauma-related pathologies in humans is limited. This is not only due to the vast species differences in diverse cognitive abilities, but because animal studies do not allow for the analysis of the complex appraisal processes and coping mechanisms that humans adopt to deal with stressors, and most often they cannot address the influence of other psychosocial factors on stress responses. Ironically, for obvious ethical reasons, the distress imposed on animals in laboratory studies comes nowhere near the trauma humans too often inflict on one another. These caveats notwithstanding, the data from animal studies have pointed to the importance of psychosocial stressors and adverse early-life experiences in the provocation of numerous long-term negative behavioral outcomes, particularly those that reflect anxiety and depression, as well as subsequent stress sensitivity and reactivity [84]. 

In humans, neurobiological changes associated with depression and forms of anxiety have been linked to adverse early-life experiences [85]. Among individuals who have experienced sexual, physical, or emotional abuse during early life, stress-related cortisol diurnal profiles are altered [86], and are predictive of disturbed executive functioning [87]. Moreover, such experiences have been associated with risk for various non-communicable diseases (e.g., heart disease, type II diabetes), as a result of elevated inflammatory processes in adulthood, reflected by increased C-reactive protein levels [88]. Effects such as these have appeared to vary with psychosocial factors and feelings of belonging to social groups. Relevant to our discussion of the mental health of Indigenous Peoples, among adults living on the Blackfeet reservation, childhood adverse experiences were related to elevated levels of a circulating inflammatory marker (i.e., C-reactive protein), occurring most prominently among individuals who self-reported low levels of belonging to the community [89].

Neuronal reactivity within the brain can be affected by adverse childhood experiences. For instance, among children that have experienced family violence and discord, the presentation of images comprising angry faces (but not sad faces) led to elevated neuronal activity in certain brain regions (the anterior cortex and amygdala), perhaps reflecting threat responses [90]. These brain changes might ordinarily be an adaptive response to a potential threat, but the magnitude of these neuronal responses points to the persistent effects of aversive experiences, which may signify elevated risk for further pathology in response to subsequent stressors. 

Too often, the adverse experiences of childhood are not isolated events but reflect a pattern of repeated psychological or physical challenges that conspire to produce mental illnesses, such as PTSD. The experience of young people living in such conditions, as has so often been apparent in Indigenous children and youth, has been described as not simply reflecting an ‘event’ but instead is a ‘condition’ in which stressors are persistently present [91]. 

### Intergenerational Consequences of Prenatal Stressors

Just as Indigenous children are more likely to encounter ACEs than those who are non-Indigenous, as noted earlier, prenatal challenges (stress experienced by pregnant mothers) are more likely to occur among Indigenous women, with implications for the fetus. In animal models, it has become increasingly apparent that prenatal stressful events may have physical and emotional repercussions that appear in offspring. Like the effects of early-life adverse experiences, it was found that stressors experienced by a pregnant dam influenced the neurotrophins that are fundamental for the neurodevelopment and memory processes, and could thereby promote behavioral disturbances [92]. In humans, stressors experienced during pregnancy may profoundly influence the well-being of offspring. The appearance of low birth weight (which is accompanied by the underdevelopment of brain and body functioning) has been apparent in response to diverse stressful experiences, including wartime conditions, domestic violence, and racial discrimination, as well as in the offspring of mothers that experienced PTSD during pregnancy [93]. Moreover, preterm birth is related to cumulative life stressor experiences, including those encountered prior to pregnancy [94]. The precipitation of preterm birth and low birth weight may occur owing to the provocation of numerous hormonal and inflammatory alterations introduced by stressors [95,96].

Individual stressor experiences (e.g., abuse) during pregnancy can have marked effects on the fetus that are manifested as far-reaching postnatal health disturbances. Developmental delays, including emotional and cognitive disturbances associated with variations of limbic and frontotemporal neural networks, have been reported in the offspring of mothers that experience emotional, psychological, physical, or sexual violence during pregnancy [97,98]. Commensurate with these findings, such maternal experiences have been linked to altered hippocampal volume and poor social–emotional development in offspring [99]. Likewise, the stress of exposure to natural disasters could affect pregnant women and lead to offspring displaying poorer cognitive, emotional, and behavioral outcomes [100]. In addition to the psychological disturbances engendered by prenatal stressors, these experiences have been associated with childhood immune-related disorders, such as allergies and asthma, and increased susceptibility to infectious diseases [101]. Stressful events experienced prenatally are similarly accompanied by elevated levels of inflammatory factors [102] that favor the subsequent development of chronic diseases, such as type 2 diabetes, heart disease, and some forms of cancer [44].

It is not simply during pregnancy that maternal stress can affect fetal development. Stressors encountered by women prior to becoming pregnant can affect offspring birth weight [103], fetal brain development, and postnatal psychological disorders [104], which could potentially come about due to the distress related to persistent negative rumination. A prospective analysis similarly revealed that preconception PTSD was associated with elevated negativity when children were 3–5 years old, even after controlling for prenatal and postnatal depressive symptoms or sociodemographic factors [105].

## 5. Epigenetic Changes Related to Stressful Events

Based on reports that traumatic events could induce intergenerational epigenetic changes among survivors of the Holocaust, as well as those experienced by other groups, it has been suggested that similar actions may affect Indigenous Peoples in Canada and elsewhere [106]. However, data are not currently available that assess this possibility. In part, this lack of research is not surprising given the numerous instances in which Indigenous Peoples have been experimented on without consent, lied to about the goals of research [107], and over-researched while at the same time remaining invisible in research [108]. Moreover, there is a concern that genetic research might be used to undermine the responsibility of colonial governments to acknowledge the causal role of history in relation to health inequities. Thus, while understanding how intergenerational trauma and environmental toxicants can influence epigenetic processes, there is currently an absence of mutually beneficial research addressing these issues among Indigenous people in Canada.

Epigenetic research reflects an understanding that social challenges, environmental agents, and even dietary factors can alter the expression of a great number of genes, without altering the genome itself. Through various processes, the functioning of particular genes can be silenced or activated, which can promote multiple phenotypes [109]. Thus, a history of colonial trauma encountered by Indigenous Peoples may have altered gene expression, leading to elevated vulnerability to psychological and physical illnesses. At one time, it had been assumed that epigenetic actions were infrequent, but it is now known that epigenetic changes are exceedingly common, with some becoming fixed (permanent) whereas others are transient. Childhood abuse, for instance, is accompanied by a great many epigenetic marker beyond those evident among individuals who have not experienced early-life abuse [110]. In view of the numerous epigenetic changes that occur, it is difficult to determine the correspondence between specific changes and particular phenotypes. Moreover, finding such relationships does not imply causality. 

The majority of the research assessing epigenetic changes has been conducted in animal models, particularly in relation to intergenerational outcomes. Studies in rodents have revealed that stressful events can promote epigenetic changes in genes that code for several hormones, neurotrophins and immune factors, which may favor the occurrence of psychological and physical disturbances [111]. Not unexpectedly, poor diet and nutrition similarly influence the expression of genes that can affect well-being. Specific nutrients (or lack of nutrients) may likewise influence the many bacterial species present within the gut and other parts of the body, and alterations of these ‘microbiota’ can produce epigenetic actions, which then promote the development of psychological disorders. Aside from nutrients, microbiota changes can be influenced by other lifestyle factors, stressors, and environmental toxicants, which can influence gene expression and their consequences.

Like other stressors, poor maternal care of pups during early development may promote epigenetic changes that have been linked to hormonal and neurotrophic processes. This includes neglect experienced by pups during early postnatal development, which can affect the epigenetic processes related to the gene regulation of glucocorticoid functioning, which is linked to elevated stressor reactivity [112]. As well, epigenetic actions stemming from early life adversity have been observed in genes associated with diverse behavioral disturbances. These include genes that influence serotonin processes [113], estrogen receptor functioning in the brain [114], as well as neurotrophin activity in several brain regions [115], all of which can affect normal behavioral functioning and the development of psychiatric disorders. In addition to epigenetic changes being provoked by stressful events during early life, similar actions were provoked in rodents stressed during the juvenile period (the equivalent of adolescence in humans); these rodents subsequently displayed increased reactivity and anxiety. Intense stressors experienced in adulthood likewise promote epigenetic effects that influence neurobiological processes linked to depressive-like features [116,117]. 

Among humans, epigenetic changes stemming from ACEs may influence health outcomes. The epigenetic changes introduced by childhood stressors are not restricted to those related to brain processes, having been seen in relation to immune cell functioning and inflammatory factors that could potentially contribute to the appearance or progression of varied diseases [118]. Of course, numerous factors, including genes coding for an array of hormones and growth factors, may contribute to later health disturbances as might an array of experiential and environmental influences (e.g., parental behaviors, poverty).

### 5.1. Epigenetic Changes Associated with Prenatal Stressor Events

Just as stressors encountered during early life can have long-term ramifications, epidemiological studies have indicated that lifestyles of women during pregnancy (e.g., diet and obesity, tobacco and alcohol use), as well as stressful experiences and exposure to air pollutants, are accompanied by epigenetic changes that could affect offspring [119,120]. Likewise, psychological stressors experienced during pregnancy can elicit epigenetic changes that influence vulnerability to pathology in offspring, and these actions could be transmitted to subsequent generations [121]. A longitudinal analysis conducted among the offspring of women who were pregnant during Hurricane Sandy in 2012 indicated that when they were 3–4 years of age, they exhibited elevated levels of anxiety and aggressive behaviors, together with increased cortisol levels measured in hair samples. These behavioral and hormonal alterations were accompanied by placental mRNA changes that were tied to endocrine and immune processes [122]. It similarly appeared that, in the offspring of women who experienced a severe ice storm while they were pregnant, cognitive disturbances were subsequently apparent that seemed to be mediated by epigenetic changes linked to genes associated with the promotion of diabetes [123]. Indeed, reprogrammed gene expression within the fetus elicited by prenatal stressors resembled those associated with early-life adverse events [124]. These epigenetic effects could involve actions related to diverse biological processes, and particular attention has been devoted to those that influence or reflect stress reactivity (HPA functioning) and emotionality in offspring [125].

While the phenotypic changes associated with prenatal stressors might have been directly related to epigenetic actions stemming from stressful experiences, they could just as well reflect poor rearing conditions and parental distress and anxiety following a traumatic experience. Based on analyses of offspring that were born following in vitro fertilization, wherein they were biologically related or unrelated to their birth mothers, it was concluded that psychological and cognitive disturbances varied as a function of genetic factors, the prenatal environment, and post-natal maternal factors [126]. In effect, the stress experienced by the biological mother may have contributed to epigenetic changes in offspring, even in the absence of prenatal or postnatal stressors. 

### 5.2. Transgenerational Transmission through Epigenetic Changes

While not diminishing the functional consequences of maternal and psychosocial factors, it has been clear from studies in animals that trauma-related epigenetic actions can affect the well-being of offspring across generations. The epigenetic changes brought about by stressors and other environmental exposures can be transmitted from parents to offspring if these alterations occur in germline cells (i.e., sperm or ova) and may occur across several generations, even if further stressors or other experience-dependent changes are not encountered [127,128]. Among other factors, exposure to stressors during pregnancy may promote sex-dependent transgenerational effects on anxiety-like behaviors, together with epigenetic changes within genes that code for or influence glucocorticoid receptor functioning and neurotrophins within particular brain regions [129,130].

Some of the transgenerational effects of the stressors discovered primarily occurred in males, whereas others were more evident in females. Sex-dependent effects have been assumed to be related to maternal influences, as mothers are the primary caregivers in most rodent species. Several studies, however, have suggested that epigenetic outcomes can occur through paternal transmission. Specifically, maternal separation from their male offspring resulted in the pups subsequently exhibiting depressive-like behaviors and increased reactivity to stressors as adults. In the offspring of these males, epigenetically related behavioral and neuroendocrine disturbances were evident, even though they had not come into contact with their male parent [131,132]. It was similarly reported that the impact of exposure to a stressor over 6 weeks prior to breeding led to paternal epigenetic effects [133]. Consistent with studies in animals, the paternal transmission of epigenetic changes stemming from ACEs in humans was linked to attention problems apparent in their offspring [134]. The early-life stressful experiences of a male parent were similarly associated with altered brain development in neonates, even after controlling for multiple maternal factors [135].

It was similarly demonstrated that early-life paternal challenges that comprised a series of different stressors could engender behavioral disturbances that were evident up to the fourth generation of offspring [136]. Effects such as these have been attributed to epigenetic actions that are especially notable among stress-reactive mice, pointing to the fundamental importance of individual difference factors in predicting the transmission of stressor effects over generations. As well, the intergenerational effects of stressors are determined by the characteristics of the stressor experienced (e.g., social versus asocial stressors; acute versus chronic stressors) and the age of the animal at the time of stressor exposure [137]. 

Predictably, far fewer studies have assessed the intergenerational and transgenerational effects of stressful experiences in humans. The length of time for each generation to come to the age of having children has limited the research regarding the intergenerational effects of trauma. Even when studies were conducted to assess the transmission of stressor effects across generations, it was uncertain whether the effects observed were related to the trauma experiences, given that they were assessed retrospectively. Moreover, most complex psychological and physical disorders are mediated by multiple genes whose actions can be moderated by numerous psychosocial and experiential factors, making attributions to epigenetic changes and intergenerational outcomes tenuous. Nonetheless, an array of factors related to stressors have been identified that are subject to epigenetic changes, some of which might be relevant to intergenerational outcomes.

Epigenetic changes in genes associated with glucocorticoid functioning have been observed in relation to aging, as well as stressor experiences [138], and it has been found that aging and stressor actions are synergistically linked in affecting pathology and inflammatory processes [139]. Adverse childhood events are similarly tied to epigenetic changes that are associated with brain changes relevant to both executive functioning and adult depression [140]. Moreover, among individuals with a history of childhood abuse and neglect who had died by suicide, epigenetic changes were present within the genes of the glucocorticoid receptors within the hippocampus [141].

The intergenerational effects of traumatic events have been examined more extensively among survivors of the Holocaust than among other groups. The survivors frequently displayed elevated glucocorticoid functioning in response to stressors [142,143,144], and such effects were frequently apparent in the adult offspring of Holocaust survivors. These behavioral and hormonal alterations were, in many instances, accompanied by epigenetic changes in both the survivors and their children [145], varying in relation to whether one or both parents had been survivors, as well as whether one or both parents had developed PTSD [146]. These data imply neither causal nor direct connections between particular epigenetic effects and any specific phenotypic changes, especially as the latter may be mediated by parental mental health problems and disturbed parenting styles [147]. Importantly, offspring might not ordinarily present with disturbed day-to-day behaviors, but the actions of epigenetic changes might be most evident upon encounters with further adverse events [148]. 

As alluded to earlier, the intergenerational epigenetic changes associated with stressors were not unique to Holocaust survivors. Many women who experienced extreme abuse (sexual violence, torture) during the Kosovo war developed PTSD during pregnancy, and epigenetic changes were detected in their children, including those that coded for a glucocorticoid receptor and a serotonin receptor, as well as a neurotrophin [149]. Paralleling these findings, the children of pregnant women who developed PTSD during the 1994 Tutsi genocide subsequently displayed increased epigenetic changes tied to glucocorticoid receptor levels relative to those apparent in women that had not been exposed to the trauma [150]. The presence of an epigenetic marker related to the oxytocin receptor has also been associated with changes in discrete brain regions related to altered empathy in mothers [151], and could affect parent–offspring interactions.

Epigenetic changes do not only occur as a result of psychosocial stressors. Environmental toxicants and drugs consumed during pregnancy may promote teratogenic effects in a developing fetus, and epigenetic variations can similarly be engendered in the fetus by environmental challenges. Endocrine-disrupting compounds, such as phthalates and bisphenol-A, as well as pesticides and dioxins, can promote epigenetic changes that are transmitted to the next generation of rodents, thereby increasing health risk [152]. Pesticides and fungicides can similarly promote epigenetic changes that increase vulnerability to viral challenges, and these actions can be transmitted transgenerationally [153,154]. Paternal intergenerational effects have been implicated with epigenetic changes associated with nutrients consumed and with deficiencies in folic acid [155]. Likewise, epigenetically transmitted actions related to paternal diet (high fat or low protein) are associated with risk of disease in offspring [156]. Given that stressors can affect nutrient intake, it can reasonably be expected that intergenerational effects can be transmitted indirectly by a male parent.

Lasting epigenetic changes have been reported among individuals that were born during severe famine conditions. Near the end of the Second World War, towns in the western parts of the Netherlands that had been occupied by Nazi forces were prevented from obtaining food or fuel, leading to misery and starvation. Children born during this period, referred to as the Dutch Hunger Winter, subsequently presented with elevated heart disease and the risk of type 2 diabetes. Psychiatric disturbances such as schizophrenia and depression occurred more often, and poor cognitive performance was apparent in later life [157]. The experience of the Dutch Hunger Winter was associated with genetic changes that influenced birth weight and subsequent LDL cholesterol levels [158]. Moreover, analyses conducted decades after this famine indicated that individuals who had prenatally experienced this condition expressed epigenetic changes related to factors that may have disposed individuals to type 2 diabetes [159,160].

Data collected in a different context are consistent with those obtained from studies of the Dutch Winter hunger. Specifically, among children born during the Great Chinese famine (1959–1961) the risk for adult type 2 diabetes was elevated and this outcome could have been transmitted across generations [161] (Li, 2017). Likewise, early-life differences in famine conditions within two regions of China revealed differences in epigenetic changes associated with neurotrophins, together with changes in cholesterol levels [162]. The intergenerational impact of the Holodomor genocide of 1932–1933, when Ukraine experienced a famine created by policies created by Stalin, is another case in point [5]. The constellation of behaviors associated with famine (or food insecurity) has been described as living in ‘survival mode’, including horror, fear, mistrust, sadness, shame, anger, stress and anxiety, decreased self-worth, stockpiling of food, reverence for food, overemphasis on food and overeating, inability to discard unneeded items, an indifference towards others, social hostility, and risky health behaviors. However, the imposition of such conditions and their implications have not been addressed in relation to Indigenous Peoples, despite the high levels of food insecurity in many Indigenous communities.

In view of the multiple epigenetic changes that have been associated with distressing events and with diverse psychological problems, as well as the influence of numerous psychosocial factors, it is premature to point to a single epigenetic alteration as being responsible for pathological outcomes stemming from adverse events. Many epigenetic actions have been identified among individuals who experienced stressful events during early life [117]. The involvement of these epigenetic changes in the intergenerational effect of stressors remains to be fully evaluated, but the findings are certainly in line with the view that actions other than, or in addition to, those associated with glucocorticoids may contribute to the persistent actions of early-life adversity. More extensive prospective analyses concerning the emergence of pathological conditions, together with whole epigenome analyses, could point to clusters of epigenetic markers relevant to the concatenation of factors that link generations of historical trauma exposure to illness occurrence. 

## 6. Reversing Epigenetic Effects: Implications for Resilience

Being subjected to collective trauma has long-term intergenerational consequences that are instigated through multiple psychosocial and neurobiological mechanisms. Nonetheless, resilience in the aftermath of such challenges does occur. It has been suggested that resilience is fostered when the survivors of natural events are able to express and anticipate collective solidarity and cohesion, and to act cooperatively to draw on collective social support resources [163]. Processes that ground people in an identity, self-affirmation, belonging, and safety [164], and are further imbued with spirituality can serve to promote a continuity of relationships across generations [165]. In effect, intergenerational resilience can be derived from the symbiotic relationship to the land, encompassing physical and non-physical elements, such as the social, emotional and spiritual aspects that provide the foundation of identity, social connectedness, and a sense of community and belonging [166,167]. 

Likewise, although some epigenetic changes can be permanent, others can vary over lifetime. For example, treating rats with particular agents affected processes that resulted in the reversal of epigenetic changes introduced by early-life stressors [168]. It was similarly observed that environmental enrichment could reverse the epigenetic changes that had occurred after separating pups from their mothers (i.e., early-life adversity in the form of maternal separation) [169,170]. As well, the epigenetic changes produced by prenatal stressors that influenced glucocorticoid receptor sensitivity and the development of anxiety-related behaviors were diminished by environmental enrichment provided during adolescence [171,172]. Like the actions of stressors, the epigenetic effects engendered by toxicants and factors that increased inflammatory factors were modifiable by exercise and nutritional changes [173]. Together, these reports indicate that the profound and lasting adverse effects of prenatal stressors and those experienced during early life do not necessarily doom offspring to negative outcomes. Resilience could still be promoted if given an opportunity to heal. In effect, the bell can be un-wrung. Taking appropriate measures to enhance postnatal environmental stimulation can attenuate some of the effects of prenatal challenges, and it is possible that interventions to diminish distress among pregnant women could be instrumental in diminishing the adverse effects on their offspring. However, to our knowledge there have not been any reports that have examined this possibility. 

Typically, discussions of epigenetics have focused on the adverse outcomes created by various stressors, but this is likely too narrow a perspective. Natural selection involves advantageous gene mutations being passed on across generations, thereby enhancing resilience and increasing the fitness of successive generations. In a similar fashion, certain epigenetic effects may persist across generations because they promote resilience, enhancing adaptations that facilitate health and survival [174]. As we know, certain phenotypes linked to mutations are selected because they have survival advantages so that preferred genes can be passed on across generations. These phenotypic characteristics evolve in small, graded steps, but they can also develop as sudden large changes. Epigenetic changes that occur owing to environmental factors may similarly provide survival benefits, thereby driving natural selection, much as genetic mutations act in this capacity [175,176]. In essence, epigenetic changes introduced by stressors may produce evolutionary benefits that can facilitate adaptive responses to diverse environmental challenges that transcend generations [177] (Jablonka and Raz, 2009). Data supporting this view are only now emerging [178] but suggest that, in conjunction with cultural, social, and interpersonal factors, biological processes might contribute to the intergenerational connections that are foundational to resilience.

The epigenetic changes that occur within a fetus may similarly serve in a preparatory capacity so that offspring are more ready to deal with threats that may be present postnatally [179]. Of course, this would largely be dependent on the nature of the postnatal environment. For instance, the epigenetic changes associated with famine conditions may prepare the fetus to deal with a lack of postnatal nutrients. However, if conditions have changed so that famine conditions no longer exist, then the nutrient conservation and limited energy use that might be tied to epigenetic factors could render the offspring more likely to develop obesity and the illnesses that this condition fosters. In effect, the influence of prenatal stressful events linked to epigenetic changes and the advantages or disadvantages that occur are dependent on the match or mismatch between the prenatal and postnatal environments [180,181]. At the same time, it is equally possible that epigenetic changes that occur within the fetus may not appear postnatally unless environmental conditions are such that these are called for [147]. In this regard, some actions of negative prenatal experiences can have beneficial actions on offspring’s ability to deal with later immunological threats. For example, among mice that encountered a prenatal bacterial or inflammatory challenge, epigenetic changes related to immune functioning appeared to provide them with enhanced protection against postnatal bacterial threats [182].

With the rapid environmental changes attributable to climate change, the psychological and biological changes that can magnify the effects of related stressors—such as food and water insecurity, the potential for socio-cultural instability, and the devastation created by (un)natural disasters that have been appearing increasingly more often—are apt to promote multiple health risks [183]. However, here, again, is where epigenetic changes might serve in an adaptive capacity. Under such environmental conditions, it is essential that biological changes occur quickly to avert exaggerated health risks, and epigenetic changes may be a way by which such adaptations can occur and be transmitted transgenerationally [184,185]. Of course, it is likely that these epigenetic changes can only go so far in diminishing the progressive consequence of climate change, and many epigenetic changes introduced by climate change may have adverse consequences [186]. 

The impact of traumatic events can be viewed through a different lens. It is inappropriate, for instance, to consider all Holocaust survivors or all IRS survivors as fitting neatly into a single package. Certainly, some survivors may carry the scars of the trauma with them throughout their lives [52]. Some survivors display psychological symptoms, such as survivor guilt, denial and mistrust, as well as intrusive thoughts, nightmares, anxiety and depression, even if these symptoms might appear at subsyndromal levels [187]. Other survivors, in contrast, may appear to have adjusted relatively well, although survivors and their children may nevertheless be more vulnerable to the negative neurobiological effects of further stressors and might thus be at an elevated risk of developing PTSD and depression (or subthreshold symptoms) [148]. In a large cohort that comprised more than 38,500 Holocaust survivors and almost 35,000 control participants, the survivors experienced many more serious illnesses, but many also lived longer [188]. It seemed that there were two sets of individuals; for the first, the Holocaust was aligned with future adverse outcomes, as might be expected in response to severe trauma, whereas the second set represented a group that was uniquely hardy, perhaps reflecting their greater resilience which had allowed them to survive the Holocaust. Whether these differences were related to genetics, epigenetics, or other psychosocial factors is uncertain. Regardless, the children of survivors probably should not be viewed as being more vulnerable to pathology. For many, these occurrences might depend on the environmental and psychosocial conditions that follow that might disrupt their capacity to cope, but others may come out of such traumatic events more resilient.

## 7. Conclusions

As alluded to at the outset of this paper, across cultures and nations, numerous groups have not only experienced efforts to eliminate them but, in some instances, genocidal efforts have recurred over centuries. The descendants of historical trauma do not view these as isolated events, but see each as chapters in a compendium of threats to their very existence. This has been apparent among Indigenous Peoples within North America, as it has among Indigenous Peoples within the U.S., Australia, and New Zealand. Trauma across generations has likewise been experienced by the Alevi Kurds within Turkey [189], the Baha ‘I’ people within Iran [190], and among Jews who have experienced physical and cultural genocide across many countries for millennia [191]. Individuals may not think about these experiences on a day-to-day basis, but reminders, including implicit and explicit ongoing challenges to their culture, language, and identity, disputed land claims, class action suits and human rights tribunals, may elicit emotional and cognitive responses that can produce a strain on biological systems. However, these experiences may also evoke self-protective responses [106], especially when group members strongly identify with their culture [192].

There is no question that the legacy of cumulative historical trauma experienced by Indigenous Peoples in Canada and elsewhere has taken an enormous toll on well-being, with the IRSs (along with the Sixties Scoop and the continued movement of Indigenous children into the child welfare system) being among the relatively recent assaults that have undermined their physical and mental health. The scarring provoked by collective historical trauma experiences may be manifested at the family and community levels, promoting profound effects on social dynamics, processes, structures and functioning. Clearly, the numerous actions perpetrated against Indigenous Peoples not only affected those who directly experienced the transgressions but continues to evoke mental and bodily harm to current group members. 

What this research has demonstrated is that the collective, historical trauma of IRSs and similar schools in the U.S. has promoted a cognitive framework that has fostered the development of diverse disturbances to well-being [193]. Adult offspring of those who have experienced traumatic events frequently have thoughts about the diverse historical losses that have been endured by their people. This includes the loss of traditional lands, the loss of language and culture, the loss of respect for traditional ways, and the loss attributed to many broken promises. Among American Indians, feelings of historical loss were accompanied by elevated blood pressure and circulating C-reactive protein, reflecting increased inflammation [194]. Moreover, as observed among First Nations people in Canada, American Indians experienced a range of cognitive and emotional disturbances. These comprised disturbed concentration, sleep problems, anxiety, and emotions such as anger, rage, shame, a fear/distrust of white people, and the avoidance of places that triggered reminders of collective losses. Much like Indigenous people in Canada, the prevalence of type 2 diabetes and heart disease was elevated by 50% among American Indians.

The Australian and Canadian governments have made efforts to cleanse themselves of the harms inflicted upon Indigenous Peoples, including offering apologies for the behaviors of their forebears. The Indigenous-led Truth and Reconciliation Commission was established in Canada to shed light on the experiences of Indigenous survivors of the Indian Residential Schools, culminating in 94 Calls to Action for the Canadian people and governments. Unfortunately, the Canadian government has been slow to address the Calls to Action and very few have been implemented [195]. Despite continued efforts to pressure the Canadian government to meet its responsibilities to address inequities and make legislative and policy changes, progress has been slow. As a result, new generations are being exposed to the continued betrayals of federal governments in meeting their responsibilities, thereby perpetuating the cycle of mistrust and resentment. 

With so many factors playing into the compromised well-being of Indigenous Peoples, multiple issues need to be addressed concurrently to bring about meaningful change. The challenges that Indigenous Peoples continue to encounter, ranging from community infrastructure (housing, clean water) to food security and quality, along with access to appropriate health services, need to be addressed now. Rather than applying singular or isolated solutions emanating from Western mental health frameworks, what is needed is to enable Indigenous Peoples to identify and implement culturally relevant strategies that reflect the connections between individuals and their social, cultural and environmental contexts. As seen in other articles in this Special Issue, approaches to wellness that address collective functioning by means of Indigenous-led initiatives that build on existing strengths and relationships are far more likely to be effective than the current Western interventions. These latter interventions typically focus on individual pathologies and, far from supporting the strengthening of cross-generational relationships, appear to be more intent on continuing the disruption of connections (and hence intergenerational trauma) by removing children so that they are raised within an alienating and underfunded child welfare system.

Western medicine has increasingly focused on the identification of biomarkers that could be used for precision (personalized) medical treatment. Among other things, these have included the identification of genetic and epigenetic factors tied to specific types of physical diseases, and efforts have been made to use this approach to address mental health disturbances. In theory, this could be achieved among Indigenous people as well but, given the lack of research (for many reasons noted earlier), the links between biological processes, health risks, and effective treatment strategies have yet to be realized. Mapping biological variations onto risks and mental health outcomes would necessarily gain the consideration of culturally defined strengths. In this regard, the protective factors that favor resilience likely vary with experiential, relational and environmental contexts, and there will be appreciable variation among Indigenous communities [196]. Accordingly, in considering a precision medicine approach to well-being, it is not sufficient to focus simply on biological mechanisms independent of historical and cultural context.

Despite continued resistance or purposeful apathy from colonial governments and organizations, Indigenous Peoples are reclaiming their inherent rights to self-determination and reconnecting to their cultural roots, including renewing land-based values, skills, and spiritual connections among the next generation of young people. Indigenous understandings of wellness are inherently holistic, with resilience emanating from the adaptive balance of protective and risk factors over time and space [197]. This may provide the perspective needed to counter the centuries of genocidal actions targeted against them. The adaptability of humans, socially, psychologically, spirituality and biologically (including through processes such as epigenetics) gives rise to hope for the future. Indigenous perseverance, despite genocidal colonialist policies, is not grounded in defeat, but in resistance. Identifying the core strengths that exist within communities, and acknowledging the complexity, contradictions and self-determination of Indigenous Peoples will be key to nurturing these roots to empower the Seventh Generation to enact an age of healing, wellness and connection.

## Figures and Tables

**Figure 1 ijerph-19-06455-f001:**
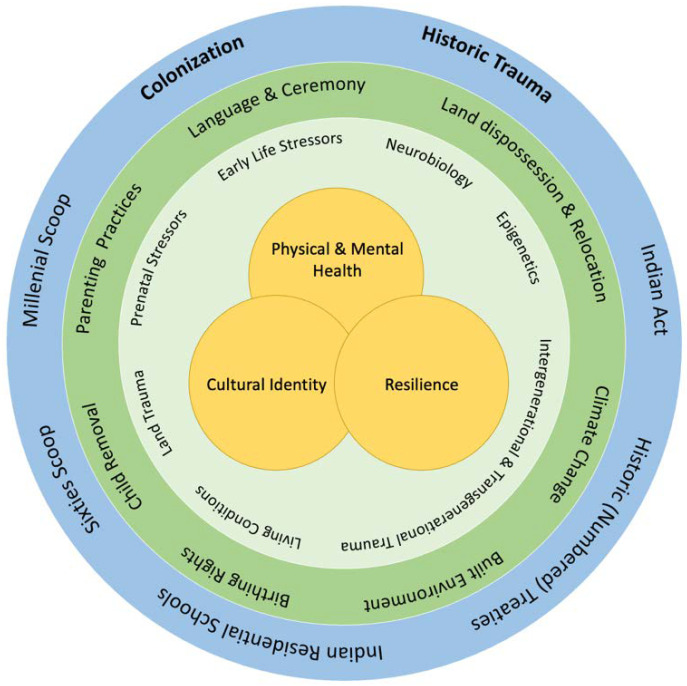
The determinants of health framework we are presenting to outline our understanding of the genocidal impacts of Canadian policies regarding Indigenous Peoples. The outer circle represents the federal policies and programs; the adjacent inner circle constitutes the targeted actions that were taken to achieve the goals of the policies, affecting the relational, cultural, and environmental context of Indigenous Peoples. As a result, these actions gave rise to processes (third inner circle) that have contributed to the health and well-being of Indigenous Peoples.

## Data Availability

Not applicable.

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
