# Peer review of "Canada’s Colonial Genocide of Indigenous Peoples: A Review of the Psychosocial and Neurobiological Processes Linking Trauma and Intergenerational Outcomes"

_ijerph, 2022, doi:10.3390/ijerph19116455_

Round 1

Reviewer 1 Report

This paper provides a historical review of the colonial oppression inflicted upon indigenous people in Canada and a review of the social, psychological, and neurobiological pathways by which these inter-generational traumas continue to impact the physical and mental health of indigenous Canadians. Overall, this is a remarkably strong manuscript that does an impressive job of reviewing a diverse set of different areas of scholarly literature with exceptional depth and clarity. I have a few points to raise, which I hope might be useful in refining it further:

  1. The only substantive respect in which I think that this paper could be improved is by adding some material to the discussion and/or conclusions regarding future potential areas for research and applied action. It is noteworthy that there seems to be little direct research on the effects of intergenerational trauma in indigenous peoples (whether in Canada or elsewhere) so this seems like a natural discussion point as a critical area for future research (it could also be discussed in terms of how this absence of research may stem from lack of funding that itself reflects a form of ongoing colonial neglect).
  2. Relatedly, it would also be informative to hear briefly how the insights gained from research in the biology of inter-generational trauma might inform programs aimed at addressing health inequalities for these groups.
  3. Given the topic, it would be appropriate to note whether the authors themselves identify as indigenous Canadians.

Author Response

1. The only substantive respect in which I think that this paper could be improved is by adding some material to the discussion and/or conclusions regarding future potential areas for research and applied action. It is noteworthy that there seems to be little direct research on the effects of intergenerational trauma in indigenous peoples (whether in Canada or elsewhere) so this seems like a natural discussion point as a critical area for future research (it could also be discussed in terms of how this absence of research may stem from lack of funding that itself reflects a form of ongoing colonial neglect).

In response to the comments of the reviewers, we have added more consideration to future research needs and implications for action to our conclusions section.  We want to note specifically in relation to this comment, at least in Canada, that funding is not the primary issue impeding research with Indigenous peoples.  We note in lines 614-627 that there are several other socio-political-historical issues that contribute to the lack of research.

2. Relatedly, it would also be informative to hear briefly how the insights gained from research in the biology of inter-generational trauma might inform programs aimed at addressing health inequalities for these groups.

While we agree that recognition of intergenerational trauma is necessary to contextualize the current situation of Indigenous Peoples, we have not taken a stance on how this is to be done.  Too often as westerners we believe that we are well-positioned to make such decisions, but in this instance, this would represent a perpetuation of sometimes well-meaning colonial perspectives. Rather, we have suggested that the disruption of the intergenerational cycle requires respecting the self-determination and leadership of indigenous peoples in the development of culturally appropriate programs.

3. Given the topic, it would be appropriate to note whether the authors themselves identify as indigenous Canadians.

We have noted in the author contributions that Seymour is Indigenous.

Reviewer 2 Report

Thank you for the opportunity to review the article “Canada’s Colonial Genocide of Indigenous Peoples: A Review of Psychosocial and Neurobiological Processes Linking Trauma and Intergenerational Outcomes.” The article is interesting and engaging, contributing to a new perspective on studying indigenous people’s transgenerational and intergenerational transmission processes, highlighting their psychosocial, developmental, environmental, and neurobiological mechanisms and their trauma responses to cultural genocide and federal policies. The authors used a literature review on intergenerational trauma “instigated by the historical genocide and ongoing genocidal conditions that shape the lives of Indigenous Peoples in Canada” (lines 97-98).

Overall, the subject is corrected attributed to the Mental Health section of the International Journal of Environmental Research and Public Health journal “, special issue “Mental Health of Indigenous Peoples,” by studying how colonialist policies and actions have affected indigenous people in Canada. The subject of the article is of interest to this field, and the perspective approached by the authors is innovative as they explore the socio-psychological implications of genocidal actions on the descendant generations and their trauma responses.

First, it is important to emphasize that the article is well-structured, the arguments are very clearly presented, and they are logically connected with the aim posed in the article's abstract.

Secondly, the theoretical background and the literature review are appropriate for this subject and the presented arguments.

I would also like to acknowledge and appreciate the presentation of the context and the state of research in the field.

However, two suggestions could be addressed in this revision (that I consider minor) which might improve the scientific soundness of the article.

  • The explanation of the framework on the third page could be better developed (lines 99-108) in order to explain why those determinants are essential and why they were chosen to explain the phenomenon. A short argument for each of the categories would be appropriate.
  • I would also propose a discussion section to identify what research could be developed to better understand these psycho-social implications in order to generate new social policies to improve the living conditions and the quality of life of the indigenous peoples.

Author Response

1. The explanation of the framework on the third page could be better developed (lines 99-108) in order to explain why those determinants are essential and why they were chosen to explain the phenomenon. A short argument for each of the categories would be appropriate.

 We have clarified the basis for our choice of determinants in our description of the model on lines 101-111.

2. I would also propose a discussion section to identify what research could be developed to better understand these psycho-social implications in order to generate new social policies to improve the living conditions and the quality of life of the indigenous peoples.

As noted in response to the comments of other reviewers, we have added greater consideration of the implications for action in our conclusions section.  We have not taken a stance on how this is to be done.  What we have indicated is that the challenges that Indigenous communities face are complex and multidimensional, and that singular or isolated solutions, particularly if grounded in western worldviews, are unlikely to be effective. Indigenous perspectives on health are holistic, and self-determination through Indigenous-led solutions needs to be supported.

Reviewer 3 Report

This is an excellent review that needs some minor changes. It is well written and very informative.

One of the weaknesses of this review is the lack of methods. A few sentences about which data bases were searched and what type of articles they assessed would set up the stage better. Furthermore, the authors should state if the figure was developed by themselves or adapted from a different publication.

Although the conclusions are very informative, it would be easier if the authors could be more concise. One suggestion is to separate this section into conclusions, and future recommendations for the Canadian government and communities since this review focused on Canadian outcomes.

Author Response

1. One of the weaknesses of this review is the lack of methods. A few sentences about which databases were searched and what type of articles they assessed would set up the stage better. Furthermore, the authors should state if the figure was developed by themselves or adapted from a different publication.

We have added to the description of our decision-making regarding the research that we covered in this review paper on lines 119-125. We also note that the figure was our own, providing an organizing framework for the issues that we address in this review and how they relate to one another.

2. Although the conclusions are very informative, it would be easier if the authors could be more concise. One suggestion is to separate this section into conclusions, and future recommendations for the Canadian government and communities since this review focused on Canadian outcomes.

We have chosen to leave the conclusions and future recommendations as a single section. We have reframed the conclusions to provide greater consideration to the implications for policy and social action in a way that is contextualized in historical and current experiences of Indigenous Peoples.